# Identification of high-risk contact areas between feral pigs and outdoor-raised pig operations in California: Implications for disease transmission in the wildlife-livestock interface

**Laura Patterson**[1,2], **Jaber Belkhiria**[2], **Beatriz Martínez-López**[2], **Alda F. A. Pires**[1]*

**1** Department of Population Health and Reproduction, University of California-Davis, Davis, California, United States of America, **2** Center for Animal Disease Modeling and Surveillance (CADMS), University of California-Davis, Davis, California, United States of America

* apires@ucdavis.edu

**Data Availability Statement:** The project protocol, including survey and cover -letter, was reviewed and determined to be exempted by the Institutional

## Abstract

The US is currently experiencing a return to raising domestic pigs outdoors, due to consumer demand for sustainably-raised animal products. A challenge in raising pigs outdoors is the possibility of these animals interacting with feral pigs and an associated risk of pathogen transmission. California has one of the largest and widest geographic distributions of feral pigs. Locations at greatest risk for increased contact between both swine populations are those regions that contain feral pig suitable habitat located near outdoor-raised domestic pigs. The main aim of this study entailed identifying potential high-risk areas of disease transmission between these two swine populations. Aims were achieved by predicting suitable feral pig habitat using Maximum Entropy (MaxEnt); mapping the spatial distribution of outdoor-raised pig operations (OPO); and identifying high-risk regions where there is overlap between feral pig suitable habitat and OPO. A MaxEnt prediction map with estimates of the relative probability of suitable feral pig habitat was built, using hunting tags as presence-only points. Predictor layers were included in variable selection steps for model building. Five variables were identified as important in predicting suitable feral pig habitat in the final model, including the annual maximum green vegetation fraction, elevation, the minimum temperature of the coldest month, precipitation of the wettest month and the coefficient of variation for seasonal precipitation. For the risk map, the final MaxEnt model was overlapped with the location of OPOs to categorize areas at greatest risk for contact between feral swine and domestic pigs raised outdoors and subsequent potential disease transmission. Since raising pigs outdoors is a remerging trend, feral pig numbers are increasing nationwide, and both groups are reservoirs for various pathogens, the contact between these two swine populations has important implications for disease transmission in the wildlife-livestock interface.

Review Board (IRB) of the University of California-Davis (No. 1180798-1. The farm location and farm ID is confidential. There was a written confidentiality agreement (in the cover-letter of this survey) stating that all provided information is kept confidential, and all data would be reported as aggregated information. Feral pig hunting tags are provided in the text as Fig1. Predictor layers used to build a MaxEnt model are publicly available online, but rasters included in the final MaxEnt model are included in KML format as supplementary files. The contact information of Institutional Review Board (IRB) of the University of California-Davis is: OFFICE OF RESEARCH IRB Administration TELEPHONE: 916 703-9151 FAX: 916 703-9160 IRBAdmin@ucdmc.ucdavis.edu 2921 Stockton Blvd, Suite 1400, Room 1429, Sacramento, CA 95817.

**Funding:** This study was supported by the Agriculture and Food Research Initiative grant no. 2019-67011-29609 / project accession no. 1019249 from the USDA National Institute of Food and Agriculture and a University of California, Davis Academic Federation grant. The funders had no role in study design, data collection and analysis, decision to publish, or preparation of the manuscript.

**Competing interests:** No competing interests or conflict of interests.

## Introduction

Although a majority of commercial swine production in the United States (US) occurs indoors with high levels of biosecurity, the US is currently experiencing a return to raising domestic pigs outdoors [1, 2]. Before the 1950s, most swine operations in the US were small-scale family farms and either a hybrid of indoor/outdoor or solely outdoor-based [1, 3]. Beginning in the 1960s, commercial swine production began transitioning to indoor systems, based on goals to increase efficiency and reduce swine disease transmission (e.g., brucellosis) as well as a public health mandate to decrease human trichinosis cases [4–7]. However, consumer demand for sustainable or pasture-raised animal products within the past few decades has revived traditional methods of raising swine outdoors or on pasture (i.e., outdoor-raised pigs, pasture-based) [1, 2, 8, 9]. While primarily considered a niche production method in the US, outdoor-raised pig operations (OPO) (e.g., commercial pork producers, backyard operations) are broadly distributed throughout California.

A challenge in raising pigs outdoors is the possibility of these animals interacting with wildlife disease reservoirs, such as feral pigs, and an associated risk of zoonotic and/or swine pathogen transmission [10–16]. Both domestic and feral pigs share the same genus and species (*Sus scrofa*) and can be reservoirs for zoonotic pathogens (e.g., swine influenza virus, Shiga toxin-producing *Escherichia coli*) [17–23]. Also, swine diseases eradicated in conventional indoor-raised herds (e.g., pseudorabies, brucellosis) have been documented in feral swine in California and contact between feral pigs and outdoor-raised swine herds is a risk factor for the reintroduction of these diseases to domestic herds in the US [6, 7, 15, 19, 24–36]. For example, a 2016 human case of brucellosis in New York state was traced to a feral pig intrusion event on a pasture-raised pig farm. *Brucella suis* was then transmitted to domestic pigs raised outdoors in 13 other states through animal sales [25, 27]. Feral pigs could also play a significant role in the transmission and maintenance of transboundary animal diseases (TAD) introduced to North America [11, 16, 28, 29]. For instance, African swine fever (ASF) is actively spreading in eastern Europe, with wild boars transmitting this devastating disease between and within countries [30]. Similarly, wild boars abet the transmission of ASF in South Korea, spreading the virus to outdoor-raised swine [31, 32]. And most recently, ASF was identified in domestic swine in the Dominican Republic, which is the closest to the US that ASF has spread in this century [33].

During the past few decades, feral pig populations have greatly increased in the US from 17 to 41 states [34–36]. California has one of the largest and widest geographic distributions of feral pigs and this invasive species has the broadest habitat range of any large mammal except humans, which is in part due to their ability to adapt to a diverse range of ecological habitats and their opportunistic omnivore diet [36–41]. Feral pig population distribution and abundance is dynamic yet has not been documented at fine spatial units less than 1km [42]. Additionally, previous presence maps reported feral pigs for an entire county, even if there had only been a single occurrence recorded countywide [38, 43, 44].

Hypothetically, an area is at higher risk of disease transmission if it is more likely to experience interactions between feral pig and domestic pigs raised outdoors, as these outdoor-based pigs can serve as a conduit for disease spread from wildlife to humans. Locations at greatest risk for increased contact between both swine populations are those regions that contain feral pig suitable habitat located near OPO, especially those OPO with relatively low levels of biosecurity [24, 26, 36, 45]. Contact between feral pigs and outdoor-raised pigs in California has been documented, as feral pigs are attracted to agricultural regions for food, water, and mates [10, 12, 29, 46–49]. There is enormous value in identifying agricultural regions with a higher probability of feral pig contact, because these areas could benefit from targeted cost-effective disease surveillance and risk-mitigation strategies to prevent disease transmission.

Predicting suitable habitat for feral pigs (i.e., likelihood of feral pig presence) in combination with spatially characterizing the distribution of OPO can provide an important tool to ascertain possible high-risk areas of contact at the feral-domestic pig interface and identify future disease spillover areas [48, 50, 51]. Species distribution modeling (SDM) methods have been widely used in ecological studies and are becoming popular for use in epidemiological investigations of disease transmission between wildlife and livestock [49, 52–54]. Maximum Entropy (MaxEnt), which is one type of SDM, allows usage of presence-only data for the species of interest (i.e., feral pigs) [55]. In combination with biologically-appropriate covariate factors, MaxEnt is able to spatially predict the probability of suitable habitat for a species for a chosen spatial unit (i.e., pixel) [56].

These two parallel trends of expanding feral pig populations and a resurgence of raising domestic swine outside has important implications for disease transmission, which could negatively impact both public health and California's agricultural industry. To the best of our knowledge, there are no maps characterizing where suitable feral pig habitat overlaps with domestic pigs raised outdoors at the farm-level in California. The overall objective of this study entailed spatially identifying potential high-risk areas of disease transmission between these two swine populations. This objective was achieved by a three-step process: 1) predicting suitable feral pig habitat in California using MaxEnt; 2) mapping the spatial distribution of OPO in California; and 3) identifying high-risk regions where there is spatial overlap between feral pig suitable habitat and OPO, as potential disease transmission areas.

## Materials and methods

This study used secondary data, does not included field work, nor animal primary data. The survey instrument and protocols were reviewed and determined to be exempted by the Institutional Review Board (IRB) of the University of California-Davis (No. 1180798–1).

### Maximum Entropy model

MaxEnt is an established SDM method that produces an output prediction map containing estimates of the relative probability of suitable habitat areas for the species of interest (i.e., feral pigs) within each pixel, using presence-only points and predictors (i.e., covariate spatial layers) [32, 47, 51, 55, 57–61]. For feral pig presence data, we obtained feral pig hunting tags from 2012–19 that were cleaned and recorded with GPS coordinates by the California Department of Fish and Wildlife (CDFW). Hunters in California are voluntarily asked to report feral pig harvest locations by submitting hunting tags to CDFW. Using hunting records for presence-points of feral pigs or wild boars has been used in previous studies [49, 62]. CDFW 2012–19 feral pig hunting tags totaled 5,148 after removing duplicates. Due to the large amount of data points, hunting tags were also manually filtered (i.e., subsampled) by year as a way to decrease the abundance of points before running models to reduce sampling bias and increase model stability, as suggested by previous analyses of MaxEnt (Fig 1) [49, 61, 63–65].

Predictor spatial layers available online, including biotic (e.g., land cover, vegetation) and abiotic (e.g., temperature, precipitation, elevation), were included in variable selection steps, and were chosen based on known feral pig behaviors, habitat, and food preferences. Table 1 displays a subset of the more than 30 predictors initially standardized and analyzed for inclusion in model building steps for MaxEnt (Table 1) [37, 47, 66–70]. For example, AVGMODIS was the annual maximum green vegetation fraction (MGVF) combined with 12 years of normalized difference vegetation index data (NDVI) and relates to food and shrub cover for feral pigs [71–73]. NDVI measures vegetation gathered by the Moderate Resolution Imaging Spectroradiometer (MODIS) as part of NASA's satellite systems. Other variables included

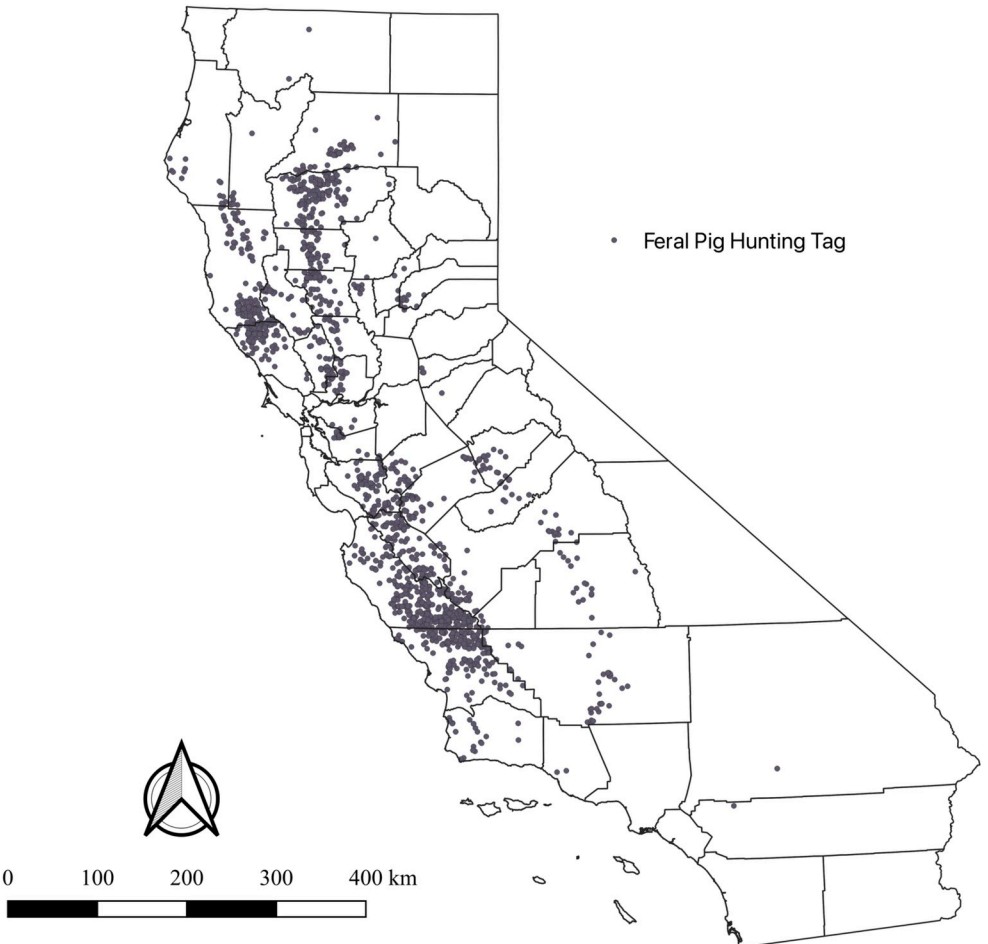

**Fig 1. California feral pig hunting tags from 2017.** Each point represents a GPS location of a feral pig hunting tag, after removing duplicate locations.

elevation, as feral pigs may prefer specific altitudes, and nineteen environmental variables from the WorldClim set of 30 year trend climatic factors [74]. Examples of environmental variables used from the WorldClim site included the minimum temperature of the coldest month (BIO6), precipitation of the wettest month (BIO13) and the coefficient of variation for seasonal precipitation (BIO15) [75]. Layers were reprojected to the Albers Equal-Area coordinate reference system for California ("California Albers" (meters)) and masked for the entire state of California, using QGIS 3.6 [76]. Rasters were all converted to the same resolution of 270m x 270m, which used a reasonable amount of computer computation time, while maintaining fine-scale for suitable habitat modeling at the farm-level. Predictors were assessed for correlation using Spearman's rank and a cut-off threshold of 0.80, a threshold used in previous studies [54]. Two correlated variables were not included at the same time, during variable selection steps.

MaxEnt models were built in R Statistical Software version 0.98.110253 © [77]. The following R packages were used to run MaxEnt: dismo, sp, and raster [58, 78–81]. MaxEnt settings were chosen based on previously published literature and included using 25 random test points, 15 replicates, 5000 maximum iterations and the 10-percentile training for the threshold rule [52, 56, 58, 82]. A regularization multiplier of one through five was assessed to avoid

**Table 1. Predictor layers assessed during variable selection for Maximum Entropy model building.**

| Name | Short Description | Year | Original Resolution | Source |
|---|---|---|---|---|
| AVGMODIS* | Annual maximum green vegetation fraction: 12 years of normalized difference vegetation index data | 2001–2012 | 250 m | modis.gsfc.nasa.gov/data/ |
| Cropland Data Layer | USDA National Agricultural Statistics Service Cropland Data | 2017 | 30 m | https://nassgeodata.gmu.edu/CropScape/ |
| Elevation* | Altitude | NA | 30 arc seconds | www.worldclim.org/ |
| FVEG | Statewide vegetation with WHR types, size, and density. | 2015 | 30 m | https://frap.fire.ca.gov/mapping/gis-data/ |
| GAP | USGS GAP analysis project: land cover | 2011 | 30 m | https://www.usgs.gov/core-science-systems/science-analytics-and-synthesis/gap/science/ |
| Global Human Influence Index | Nine global data layers: human population pressure, human land use and infrastructure, and human access | 1995–2004 | 1 km | https://sedac.ciesin.columbia.edu/data/set/wildareas-v2-human-influence-index-geographic/maps |
| NLCD | National Land Cover Database | 2016 | 30 m | https://www.mrlc.gov/data |
| Open Water | Multiple integrated global remote sensing-derived land-cover products and prevalence of 12 land-cover classes | 2005–2006 | 1 km | https://www.earthenv.org/landcover |
| PRISM | Seven climatic variables for the US annual and monthly precipitation and temperature | 2016 | 270 m | http://www.prism.oregonstate.edu/ |
| Streams | Distance to water | 2003 | NA- Shapefile | https://catalog.data.gov/dataset/cdfg-100k-streams-2003 |
| USDA zones | Hardiness zones based on mean extreme annual minimum temperatures | 2012 | NA- Shapefile | https://planthardiness.ars.usda.gov/ |
| WorldClim* | 19 bioclimatic variables: 30-year averages 1970–2000 | 1970–2000 | 30 arc seconds | www.worldclim.org/bioclim |

*Indicates variables included in the final MaxEnt model.

overfitting and the default one was determined to be the optimal setting for the final model [82]. Logistic values for output were used as well as cross validation, which separates presence points into 80% training and 20% testing data (i.e., model validation), using k-fold sub-sampling to fit a model [52, 56, 83]. The relative contribution of each variable in a MaxEnt model was assessed comparing both percent contribution and permutation of importance, averaged over the number of iterations run and ascertained by jackknife tests [52, 55, 83]. Predictors for the final model were assessed using a backward variable selection approach: variables remained at each step if their percent contribution or permutation importance were approximately 10% or more [32, 52]. The response curves generated within MaxEnt showed the predicted probability of suitable feral pig habitat for each individual variable, changing per each level of the predictor [66, 69, 84].

MaxEnt model performance was assessed using the area under the curve (AUC) of the receiver operator characteristic (ROC), averaged over the number of chosen replicate runs [52, 54, 85]. AUC reflects a model's prediction ability, on a scale of 0 to 1.00, with 0.50 representing random chance. In general, the following guidelines are used to assess AUC: 0.90 and above indicates an excellent model, a good model ranges from 0.80–0.90, a fair model runs between 0.70–0.80 and poor or failed model is any value under 0.70 [47, 56, 86]. While AUC is a standard diagnostic method to evaluate MaxEnt models, some authors suggest calibrating the AUC (i.e., AUCc), which removes spatial sorting bias (ssb) (i.e., spatial autocorrelation) by using point-wise distance sampling [58, 78, 87]. A ssb close to 1 indicates no spatial sorting bias, whereas a ssb close to 0 suggests a large spatial bias, and the need to use AUCc [87]. The final model was chosen based on the highest AUCc, relative to other models.

## Risk map and OPO

A risk map that categorized areas at greatest risk for contact between these two swine groups was built by overlapping California OPO locations with the final MaxEnt feral pig suitable habitat raster. Between 2014–2019, OPO in California were compiled through various sources (e.g., agricultural festivals, local farmers markets, University of California Cooperative Extension (UCCE) advisors, web-based searches (search terms: "pasture-raised pork", "pastured pigs"). GPS coordinates for all OPO were identified using Google Earth Pro v7.3.3. [88] Additionally as part of our objective to gather locations of OPOs in California, an online survey that contained an interactive map component was built with Survey 123 v3.6 [89]. The survey contained 29 questions that consisted mainly of multiple choice questions, with a few open ended questions about the number of animals raised (e.g., how many sows or boars raised on average each year). The survey included questions regarding biosecurity practices, swine health and feral pig presence. This online survey was announced electronically (e.g., media, e-newsletters) to swine related groups and organizations or conducted in-person at events, such as agricultural fairs. The survey instrument and protocols were reviewed and exempted by the Institutional Review Board (IRB) of the University of California-Davis (No. 1180798–1) (S1 Appendix).

To build the risk map for California, the final MaxEnt model predicting suitable habitat for feral pigs was overlapped with the location of OPO to categorize areas at greatest risk for disease transmission, due to contact between these two swine populations, and characterize risk at the farm-level. The underlying assumption presumed that direct or indirect contact between feral pigs and domestic pigs raised outdoors is a risk for disease transmission [90]. The probability of suitable habitat for feral pigs was extracted from the final MaxEnt model for each OPO location, using the Sample Raster Value tool in QGIS and added to the OPO shapefile. Then the Kernel Density tool in QGIS was used to make the risk map, matching the 270m x 270m resolution of the MaxEnt model and using the MaxEnt model probabilities as weights. Additionally, we used a radius of 5 km at each OPO location, which was an extrapolated average estimate from US based studies that measured home range of feral pigs, understanding that home ranges vary depending on age and sex of animal, as well as resource availability [47, 68, 91, 92]. The Kernel Density map was overlaid with the final MaxEnt model.

# Results

## MaxEnt model

The final MaxEnt model was chosen based on the highest AUCc of 89.7, relative to other models. Probability values for suitable habitat were divided into five equal interval categories: minimal (< 0.01); low (0.01–0.22); moderate (0.23–0.43); high (0.44–0.65); and extremely high (0.66–0.87), with 0.87 being the highest predicted probability in the final MaxEnt model (Fig 2). Areas with the highest likelihood of suitable feral pig habitat in California (i.e., orange, and red categories) included the north coast from Mendocino County all the way south along the coast to Santa Barbara County, and counties that border these coastal counties (e.g., Lake, Napa, Contra Coast, Santa Clara, and San Benito). Additionally, suitable habitat areas included the foothills of the Sierra mountains, from Shasta County south to Tulare County. Least likely suitable habitat included the Central Valley and eastern counties of California, from the most northern county of Modoc all the way to Imperial County in the south.

Five variables were identified as significant in predicting suitable feral pig habitat. The five significant variables were the annual maximum green vegetation fraction (AVGMODIS), the minimum temperature of the coldest month (BIO6), precipitation of the wettest month

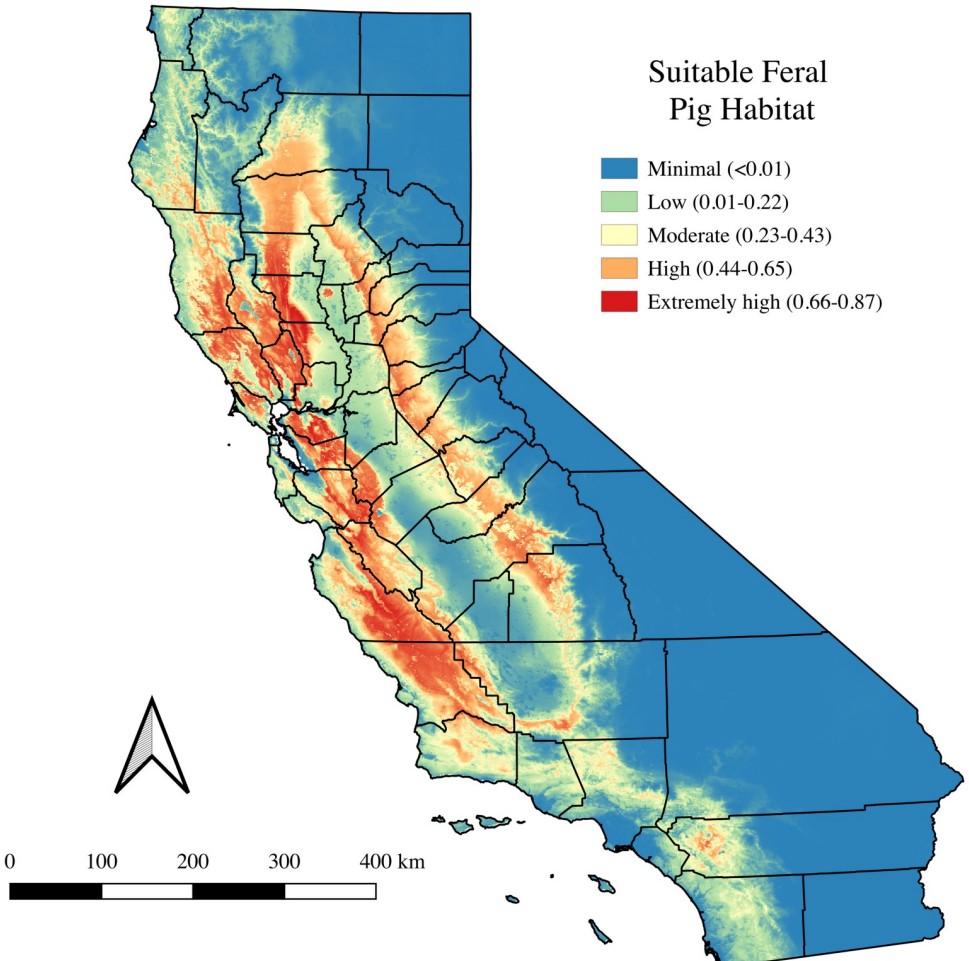

**Fig 2. Final MaxEnt model predicting suitable feral pig habitat in California.** Color-coded categories represent the probability of suitable feral pig habit on a scale of almost zero (<0.01) to extremely high (0.66–0.87), based on equal intervals.

(BIO13) and the coefficient of variation for seasonal precipitation (BIO15) and elevation. All five variables provided approximately 10% or more percent contribution and permutation importance to the final model (S1 Table). The jackknife test results provided more information regarding the importance of each variable in the final model (S1 Fig). For example, BIO15 was the variable with the highest gain when used alone and elevation had the most information that was not available in the other variables. The response curves for the significant five variables indicated the predicted suitability range of each variable for feral pigs (i.e., the x-axis values above 0.50 on the y-axis) (Fig 3). For instance, feral pigs are predicted to prefer vegetative cover (i.e., AVGMODIS) of at least 60% or more.

## Risk map and OPO

A total of 305 OPOs were identified between 2014–2019, from 79.30% (46/58) of California's 58 counties (i.e., no OPO data for 12 counties). The most OPO were identified in the following counties: Sonoma (n = 48), Mendocino (n = 19), Nevada (n = 16) and Yolo (n = 12). From the online survey, 39 OPO locations were gathered from 44 respondents and included in the final total. All survey respondents raised domestic swine outdoors and 25.00% (11/44) had seen

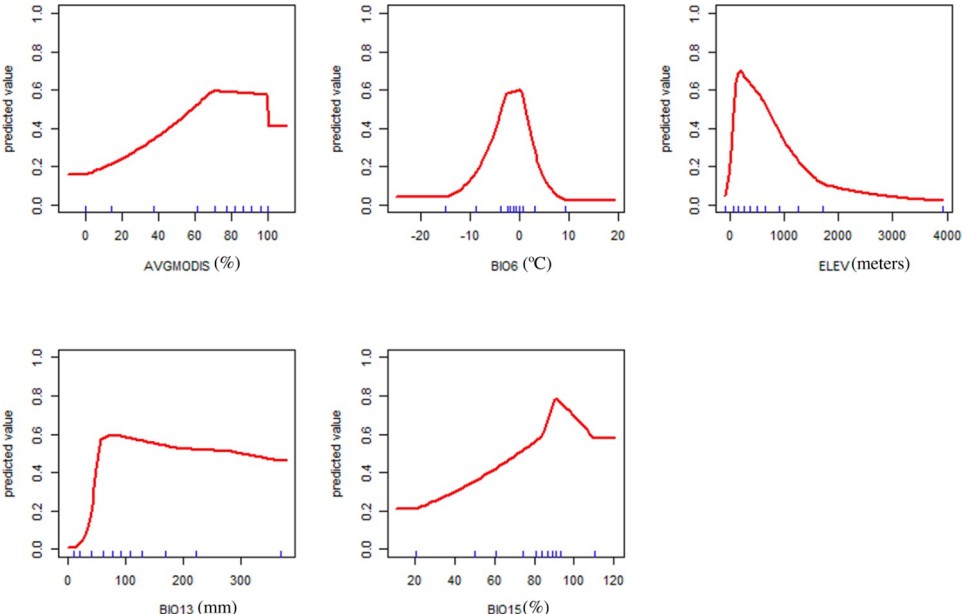

**Fig 3. MaxEnt response curves for the five significant variables used in the final MaxEnt model.** The response curves generated by MaxEnt show the predicted probability of suitable feral pig habitat for each individual variable per each level of the predictor. Significant layers included the minimum temperature of the coldest month (BIO6), the annual maximum green vegetation fraction (AVGMODIS), the precipitation of the wettest month (BIO13), the variation of annual precipitation (BIO15) and elevation.

feral pig presence within 3.22 km or less of their domestic swine raised outdoors, with 15.91% (7/44) observing feral pigs within 152.4 m of their pig herd. Domestic pig herd size ranged from 1–350 animals, with a mean of 24 and median of six. Hectares (ha) dedicated to raising pigs ranged from 0.026 ha to 12.14 ha with a mean of 1.89 ha and median of two 0.81 ha, with nine not answering.

The risk map reflects areas at greatest risk for contact between feral swine and domestic pigs raised outdoor and subsequent potential disease transmission (Fig 4). Risk levels start at green for low-risk areas and range up to orange and red for the highest risk areas. Areas with the most risk for contact between these two swine populations are denoted in orange or red, with sharper colors representing denser clustering of OPO. The counties with the highest likelihood of suitable feral pig habitat and densest clustering of OPO included: Sonoma, Marin, Napa, Yolo, Nevada, Mendocino, and Lake counties. Areas at lowest risk include the full eastern edge of California, which includes the Cascadian and Sierra Nevada Mountain ranges as well as deserts in the south. Table 2 categorizes the distribution of OPO at each level of probable suitable feral pig habitat using the final MaxEnt model levels. The results indicated that 49.18% of the 305 OPO were located near extremely high or highly suitable feral pig habitat.

## Discussion

In this study, we investigated the predicted distribution of feral pigs in California and their spatial overlap with domestic pigs raised outdoors, to determine areas for surveillance in the case of an emerging or reemerging disease outbreak. The MaxEnt model results indicated heterogenous feral pig suitable habitat in each California county, instead of a homogenous distribution, as suggested by past maps. Additionally, this study overlapped predicted suitable feral pig habitat and OPOs to create a risk map for potential disease transmission at the feral pig-

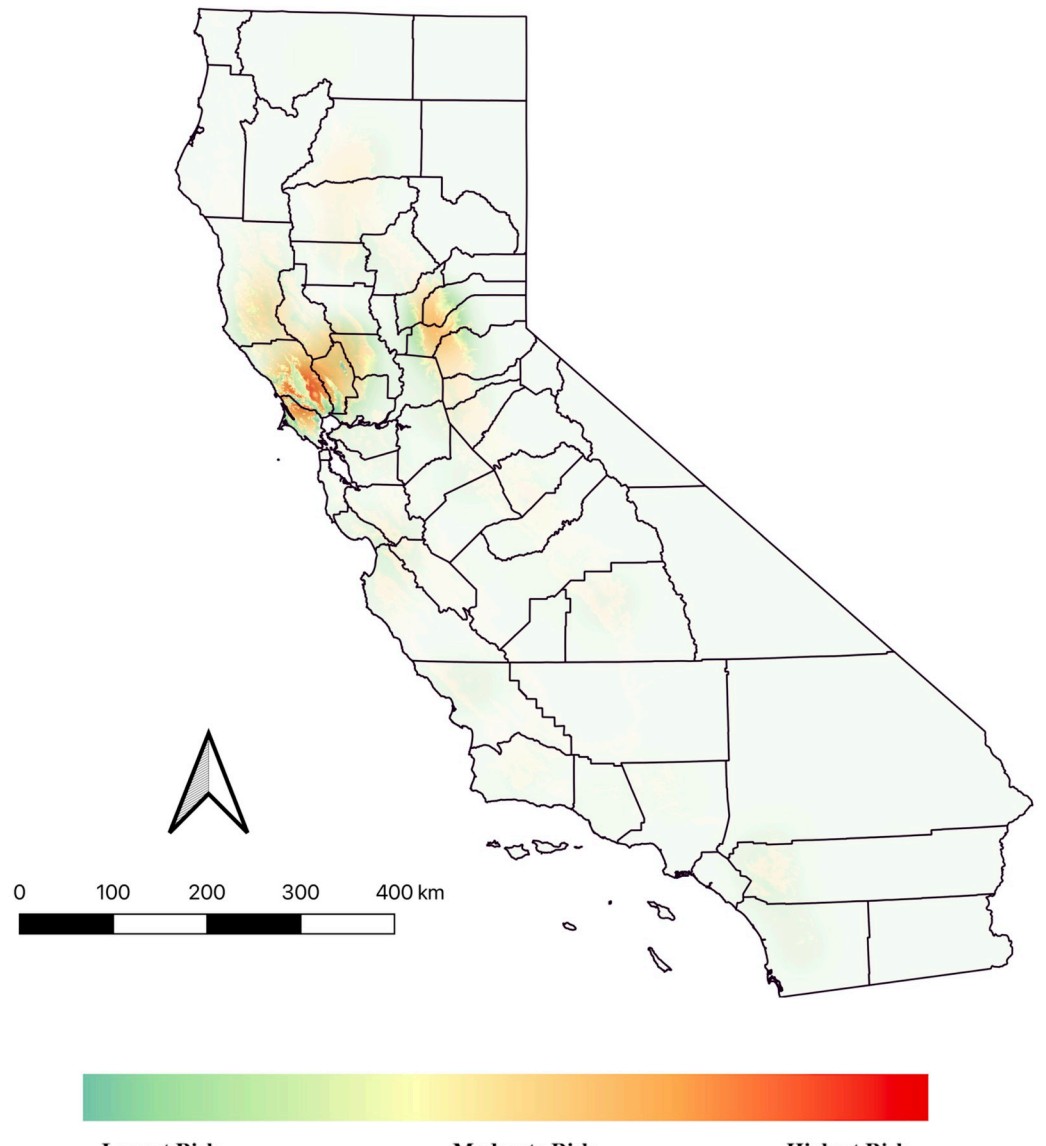

**Fig 4. Risk map demonstrating areas in California at greatest risk for contact between feral pigs and outdoor-raised domestic pigs within a 5km radius from each farm, using the Kernel Density tool in QGIS.** Colors are based on the probability of suitable feral pig habitat from the final MaxEnt model at each OPO, with sharper colors representing denser clustering of OPO.

**Table 2. Percentage of 305 OPO identified in each MaxEnt suitable feral pig habitat level.** The final MaxEnt model contains a probability scale of 0.00 to 0.87 and was divided into equal intervals.

| Levels | %OPO (ct/305) |
|---|---|
| Minimal ($< 0.01$) | 0.98% (3/305) |
| Low (0.01–0.22) | 19.67% (60/305) |
| Moderate (0.23–0.43) | 30.16% (92/305) |
| High (0.44–0.65) | 25.90% (79/305) |
| Extremely high (0.65 +) | 23.28% (71/305) |

domestic pig interface. Although previous studies discussed the possibility of feral pig populations spreading disease to outdoor-raised pigs at the county-level, this is the first study to predict risk at the farm-level in California.

Since the exact location of most feral pig populations is unknown, predictive methods for species distribution (e.g., MaxEnt) are important to understand where feral pigs could potentially interface with domestic swine raised outside, either currently or in the future. Our final prediction model provided a more informative picture of suitable habitat for feral pigs than previous studies, which only showed single presence points or reported feral pigs at the county-level, even if only one feral pig was identified in that county [36, 38, 43, 44]. For instance, although previous county-level maps stated that all California counties except for Imperial County harbored feral pigs, our model shows almost no suitable habitat in an additional five counties: Modoc, Mono, Alpine, Lassen and Inyo. This result may reflect that few feral pigs have been seen in those counties. Additionally, our results were based on a fine spatial scale and indicated heterogenous suitable habitat in counties, not a uniform distribution, which is compatible with the fact that feral pigs need shrub cover and food to survive, which would not be found in cities or deserts [41]. Earlier feral pig mapping studies by the Southeastern Cooperative Wildlife Disease Study and National Feral Swine Program (NFSP) focused on county-level occurrence in the US [43, 44, 93]. A 2015 United States Department of Agriculture (USDA) study overlapped NFSP county-based feral pig locations with data from the 2012 National Animal Health Monitoring System (NAHMS) report of small-enterprise swine operations, specifically whether these survey respondents had seen feral pigs on their premises or within the same county, to ascertain the level of agreement between the two datasets [36]. They identified five counties in California that were in agreement with our model findings for suitable feral pig habitat: Mendocino, Tehama, Nevada, El Dorado, and San Luis Obispo, and two counties that differed: Ventura and Los Angeles counties. Although these county-based maps are important to demonstrate the trend of increasing feral pig populations nationwide, stakeholders and feral pig disease surveillance agencies could benefit from targeting outreach and mitigation strategies to specific regions within a county using our maps.

The results of our final model indicated five variables that were useful in predicting suitable feral pig areas in California, including three WorldClim layers: the minimum temperature of the coldest month (BIO6), precipitation of the wettest month (BIO13) and the coefficient of variation for seasonal precipitation (BIO15). Other studies also used WorldClim factors to predict the distribution of wild boar or feral pigs. These bioclimatic variables have been widely used in environmental studies and are now becoming popular for use in epidemiological investigations [75]. These climate variables are 30 year averages and "capture broader biological trends better than the temperature or the amount of precipitation for a given day due to the inherent variability associated with weather." [75]. Bosch *et al* (2014) built a MaxEnt model for wild boar in Spain and their model also contained BIO6 and BIO15 as did regional models built by Pittiglio *et al* (2018) with BIO13 being significant as well [70, 86]. BIO6 is the minimum temperature of the coldest month and is interpreted as being a useful variable when deciding if the species of interest is affected by extreme cold events throughout a year [75]. Hill *et al* (2014) used MaxEnt to predict the distribution of *Trichinella* spp. and *Toxoplasma gondii* in feral pigs in the US and also identified BIO6 and elevation as significant predictor variables, along with land cover and other WorldClim factors [26]. The response curve for BIO6 in our model peaks at the predicted ideal range for feral pigs, with both ends indicating extreme cold temperatures that may be avoided by feral pigs. A 2015 study by McClure *et al* (2015) indicated that suitable feral pig habitat may be limited by cold temperatures, precipitation, and water availability, which reflects our findings [47]. BIO13 is defined as precipitation of the wettest month and is useful if extreme rainfall patterns influence the range of feral pigs [75]. BIO15

measures the variation in annual precipitation totals per month (i.e., seasonality of precipitation) and reflects the variability of rainfall that may affect a species [75]. According to the Jackknife graph, the variable with the highest gain when used alone was BIO15, and therefore had the most important information for predicting suitable feral pig habitat. Snow *et al* (2017) used Bayesian methods to predict the expansion of feral pigs in the US, but also detected that temperature and precipitation levels were significant predictors [38]. The final model gain is decreased the most if elevation is ignored and therefore it has significant information that is not available from the other variables in predicting feral pig suitability. Elevation was also significant in the MaxEnt models built by Hill *et al* (2014) [26]. These results combined with the response curve possibly reflect feral pigs preference for lower altitudes in the US. AVGMODIS, a measure of the annual maximum green vegetation fraction on a scale of 0 to 100, was also an important predictor of suitability, which reflects feral pigs' need for available food and vegetative cover [72]. Garza *et al* (2018) identified NDVI, which AVGMODIS is based upon, and precipitation as important variables in predicting home ranges of feral pigs or wild boar worldwide, using generalized linear models [66].

The significant layers identified in our study to predict feral pig suitability are not unique, and this may be due to the fact that feral pigs are highly adaptable and opportunistic omnivores [38]. Lobe *et al* (2008) stated that MaxEnt AUC values will be lower for generalist species that are widely distributed [94]. However, the AUCc of our final model was 89.7, which indicates a good model. Another factor to consider when analyzing the distribution of feral pigs, is the effect of anthropogenic movement. Tabak *et al* (2017) analyzed anthropogenic factors that might affect the expansion of feral pigs, using hierarchical Bayesian models [13]. They determined that feral swine movement was affected by similar factors included in our study, for instance, the number of domestic pig farms, the amount of public land and hunter pressure [13]. Although they did not specifically identify domestic swine raised outdoors, they found that presence of a domestic pig farm did predict movement of feral pigs into California counties, since domestic pigs are known to escape and can readily adapt [13]. Additionally, using a model fitted with 2017 hunting tags (n = 1,745) vs. all 5,148 provided the best prediction model, based on the AUCc. MaxEnt is an important method to predict the distribution of rare species, and an upper maximum range for the number of species occurrence points has not been previously determined. However, our result fits with a study conducted by Chen *et al* (2012) to determine the sample size for the outcome variable in building MaxEnt models [63]. They reported that standard deviation decreased and MaxEnt models became more stable using species occurrence points of 1,000–1,200 [63–65]. Possibly the sample size of the outcome variable that reaches asymptote is dependent on the geographic extent and characteristics of the species of interest.

Regarding feral pig presence on farms, the most recent NAHMS survey asked participating swine small-enterprise producers in the US (i.e., those raising 100 pigs or less) about presence of feral swine in their county, but did not separate farms based on whether they raised domestic swine indoors or outdoors. However, a 2015 USDA report regarding overlap of feral and domestic pigs in the US used this NAHMS dataset and reported that of the 320 participating US counties, 74% of these counties had small-enterprise swine producers who allowed their pigs some level of outdoor access [36]. The NAMHS results indicated that 52.9% of small-enterprise swine producers in the West/South region, which included California, reported feral pigs in the same county, with 16.2% of those having feral pig presence on their operation, similar to our survey results that showed 15.91% of respondents had seen feral pigs within 500 feet of their pig herd [95]. Another study that measured the co-occurrence of feral pigs and agriculture to understand the risk of disease transmission, but that did not separate outdoor versus indoor herds, reported that on average 47.7% of all types of farms had feral pigs in the

same counties, including California, showing a significant increase in the decade from 2002–2012, which aligns with the fact that feral pigs numbers are increasing nationwide [96, 97]. Additionally, our risk map identified eastern counties as having the lowest risk. However, we did not identify OPO in many of these counties, therefore we cannot say there is no or low risk in those regions. The results of these aforementioned survey-based studies indicated that more than 45% of farms have feral pig presence within the same county in areas with a large population of feral pigs, which matches the results from our risk map that showed almost half of the identified OPO in California had suitable feral pig habitat nearby [36, 95, 96]. Targeting outreach and surveillance to highly connected farms may be warranted, even if that farm is not near highly suitable feral pig habitat, given their effect on other farms. Additionally, OPO with the highest risk of disease spillover may not be connected to other OPOs. Nevertheless, these findings together indicate the need for targeted outreach and mitigation strategies for those farms at highest-risk for feral pig contact, due to the potential for disease transmission between these two swine groups.

The results of our risk map indicated that 49.18% of the 305 OPO identified in California are located near highly suitable feral pig habitat, indicating that spillover of an emerging or transboundary disease is a possibility, given the correct drivers. Possible drivers of disease spillover in the US between feral pigs and domestic swine raised outdoor include the density of animals (both feral and domestic swine), shared natural areas and increasing contact between these two growing swine populations [98, 99]. Although spatial overlap of these two swine populations does not necessarily demonstrate direct contact, direct or indirect contact between a pathogen host and susceptible individuals is one factor that facilitates disease transmission [90]. A Spanish study by Kukielka *et al* (2013) used camera traps to measure interactions between wild boar and domestic swine, to understand the transmission of *Mycobacterium tuberculosis* [100]. Interactions between wild boar and domestic swine were mainly recorded as indirect contact at water sites during wet seasons. Additionally, domestic pigs followed wild boar most often, instead of vice versa, and the authors suggested the spread of tuberculosis would occur mainly from exposure of domestic pigs to wild boar, through indirect contact. A study by Yang *et al* (2021), used GPS collars to quantify direct and indirect contact rates between feral pigs and cattle in Florida, a state with a high abundance of feral pigs like California [90]. While they found that direct contact was infrequent, indirect contact at water troughs or mineral blocks was significant, indicating that contact between feral pigs and livestock can contribute to pathogen transmission, assuming other disease transmission requirements are also met [90]. These aforementioned studies demonstrate the continued need to study transmission dynamics between feral pigs and domestic swine through direct and indirect interactions, as contact between feral and OPOs is possible. Other components of disease transmission that affect spillover include the density of the pathogen host (i.e., feral pigs), the number of domestic swine raised outdoors on each farm and the connections between OPO [90]. Since the exact number and location of feral pigs is currently unknown in California, using camera traps to estimate abundance might be one way to improve this current study. Additionally, we were unable to gather the number of acres and swine raised on each OPO, as this information is not readily available in the US. Regarding connections between OPO, unless a farm has a closed herd, one might assume that connections between OPO exist if the owners share tools or sell animals between farms. However, these dynamics were not measured in this study. A network analysis of connections between OPO would add more dimensionality to the current risk map.

Our risk map reflected the heterogeneity of feral pig habitat in each region and identified high-risk contact areas between farms and feral pigs in California. Studies that identified high-risk areas in California between feral pigs and domestic swine raised outdoors are sparse. A

2015 USDA report extracted outdoor operations with NFSP feral swine populations and did not identify any hot spots of overlap in California as seen in our results [36]. However, they did not report the number of OPO per state or county and most likely our state-focused study identified more OPO than their survey-based national study. A study by Miller *et al* (2017) also assessed possible disease transmission between feral pigs and farm at the county-level [96]. They reported that domestic swine, either raised indoors or outside, have been increasing in counties that also had feral pig presence. The lack of maps identifying areas at high-risk for disease transmission between these two swine populations indicates a need for further research.

A limitation of this study involves using hunting tags as a proxy for presence of feral pigs to predict suitable habitat. Hunting tags are voluntarily submitted to CDFW by hunters and estimated to account for only 30% of all hunted pigs and most likely biased toward easy to access areas. Also, only half of the land in California is public land and accessible to hunters, therefore feral pigs hunted on private land are not included in our data sets. However, Rutten *et al* (2019) used similar hunting bags and MaxEnt to successfully predict the distribution of wild boar in Belgium [49]. And Alexander *et al* (2016) also used hunting records to predict wild boar habitat in Europe [62]. Additionally, MaxEnt assists in overcoming these challenges by identifying similar habitats in all parts of California and predicting suitable areas.

Both the MaxEnt model and risk map are limited because they are static maps that use fixed layers as their foundation; consequently, they do not incorporate dynamic events over various years (e.g., wildfires, landscape changes, weather fluctuations). Also, feral pigs may migrate seasonally due to shifting weather, resource availability, hunting pressure or wildfire and future research could focus on species distribution modeling that includes dynamic real time variables or remote sensing data; however, seasonal or dynamic spatial data are not available yet for most spatial predictors in California [47, 49, 66, 91, 101–103]. However, our approach is valuable as a first step in identifying multiple high-risk areas for future research, where additional data could be collected. Furthermore, future research could add feral pig disease data collected statewide to evaluate if high-risk areas for feral-domestic pig contact equates to those areas with higher prevalence of diseases [50].

There are some challenges and limitations to the risk map generated in this study. For instance, farms and ranches in California, including backyard and commercial operations, are not required to register with state agricultural agencies, therefore, the total number, distribution, and size of OPO remains unknown and are underrepresented in this study. A majority of the identified OPO in this study were commercial pork producers with an online presence or ones that attend conferences, farmers markets and fairs. If more OPO locations could be identified, than a more comprehensive map of high-risk areas could be generated. Additionally, because we are based at the University of California, Davis in Yolo County, there is selection bias in the OPO identified as our agricultural networks are within the UCCE network. Overrepresented counties reflected either sampling bias or clustering of these niche operations or both. Nevertheless, the number of OPO included in this study (n = 305) and the fact that more than 40% of these operations were in highly suitable areas for feral pig contact is relevant as an initial approximation of a likely much larger risk of disease transmission at the feral-domestic swine interface in California. In the future, adding disease cases to this risk map would add additional epidemiological information regarding possible pathogen transmission.

## Conclusion

This study evaluated the feral-domestic pig interface of two parallel trends: expanding feral pig populations and an increase in outdoor-raised pig operations in California, as related to the

risk for future disease transmission. Since both swine populations are reservoirs for various pathogens, the contact between these two swine groups has important implications for disease transmission in the wildlife-livestock interface. This study provides a foundation to design targeted, cost-effective disease surveillance and risk mitigation programs in regions at highest risk for wild-domestic pig contact and can serve as a template for similar efforts nationwide. Moreover, the results of this study provide a framework to create an outreach extension program and inform all stakeholders (e.g., farmers, government agencies) that may be called upon to respond to future zoonotic or TAD outbreaks, such as ASF. The results of this study, despite limitations, can provide important information to stakeholders and organizations that handle swine diseases or public health problems originating from any swine group in California.

## Supporting information

**S1 Appendix. Survey for outdoor-raised pig owners in California.**
(PDF)

**S1 Table. The analysis of variable contribution table provided estimates of the relative contribution of each variable to the final MaxEnt model.**
(TIF)

**S1 Fig. Jackknife results for final MaxEnt model.** The Jackknife graph indicated importance of key variables: BIO6 was the minimum temperature of the coldest month, AVGMODIS was the annual maximum green vegetation fraction, BIO13 was the precipitation of the wettest month, BIO15 was the variation of annual precipitation and elevation.
(TIF)

**S1 File.**
(ZIP)

## Acknowledgments

The authors would like to thank the following people and organizations for their support and assistance: Robert Hijmans for Maximum Entropy expertise; Michele Tobias at UC Davis for QGIS assistance; Robert Johnson and Shane Feirer at UC ANR for building the online survey; Kristi Cripe and Craig Stowers at CDFW for hunting tag data and feral pigs information; Jim Thorne and Michele Stern for PRISM data; Jasmine Torres for online farm searches and big thanks to our feral pig experts: Dana Nelson, Rebecca Milhalco, Hector Webster, John Harper, and Roger Baldwin.

## Author Contributions

**Conceptualization:** Laura Patterson, Beatriz Martínez-López, Alda F. A. Pires.

**Data curation:** Laura Patterson.

**Formal analysis:** Laura Patterson.

**Funding acquisition:** Laura Patterson, Alda F. A. Pires.

**Methodology:** Laura Patterson, Jaber Belkhiria, Beatriz Martínez-López.

**Project administration:** Laura Patterson, Alda F. A. Pires.

**Resources:** Jaber Belkhiria, Beatriz Martínez-López, Alda F. A. Pires.

**Supervision:** Jaber Belkhiria, Beatriz Martínez-López, Alda F. A. Pires.

**Validation:** Laura Patterson.

**Visualization:** Laura Patterson.

**Writing – original draft:** Laura Patterson.

**Writing – review & editing:** Laura Patterson, Jaber Belkhiria, Beatriz Martínez-López, Alda F. A. Pires.

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
