## [Decision Letter · Decision Letter 0]

18 Jan 2022

PONE-D-21-36324Identification of high-risk contact areas between feral pigs and outdoor-raised pig operations in California: implications for disease transmission in the wildlife-livestock interfacePLOS ONE

Dear Dr. Pires,

Thank you for submitting your manuscript to PLOS ONE. After careful consideration, we feel that it has merit but does not fully meet PLOS ONE’s publication criteria as it currently stands. Therefore, we invite you to submit a revised version of the manuscript that addresses the points raised during the review process.

I agree with the comments by Reviewer 1 but I believe that Major Revision is more appropriate. I don't believe the level of detail of the dataset and spatial layers is adequate. In their revision I would appreciate the authors including details for their methods outlined below:

1. As mentioned by Reviewer 1, the level of detail of how each spatial layer was presented into MaxEnt is not adequate. I needed to see Figure 1 before realizing that this was throughout the state of California and not a portion of it. Table 1 could include resolution of each layer and how each layer was presented. The level of detail in Description column is not nearly enough. I would suggest moving this to the Methods text and add more details. I don't believe anyone could replicate your models with this level of detail for your spatial layers.

2. Was AVGMODIS an average of 12 years of NDVI.

3. If AVGMODIS is NDVI is it correlated to NDVI? Were any correlations included in your data sources prior to using them in MaxEnt?

4. What year was NLCD used as a spatial layer? NLCD is released every 5 years so which version should be placed in the methods.

5. As requested by Reviewer 1, feral pig tag locations across the state needs to be included if the authors are presenting a risk map statewide.

6. Was the OPO survey done as part of this study? The response rate of these surveys is important to report. Do the authors believe their survey adequately reflects OPOs (13 percent response rate) in California based on their results? It seems the survey questions should be a supplement or a separate manuscript? If the authors have coordinates for OPOs but only a 13% response rate, is the survey of any value in this manuscript?

We look forward to receiving your revised manuscript.

Kind regards,

W. David Walter, Ph.D.

Academic Editor

PLOS ONE

Journal Requirements:

4. We note that Figures 1 and 2 in your submission contain map images which may be copyrighted. All PLOS content is published under the Creative Commons Attribution License (CC BY 4.0), which means that the manuscript, images, and Supporting Information files will be freely available online, and any third party is permitted to access, download, copy, distribute, and use these materials in any way, even commercially, with proper attribution. For these reasons, we cannot publish previously copyrighted maps or satellite images created using proprietary data, such as Google software (Google Maps, Street View, and Earth). For more information, see our copyright guidelines: http://journals.plos.org/plosone/s/licenses-and-copyright.

 a. You may seek permission from the original copyright holder of Figures 1 and 2 to publish the content specifically under the CC BY 4.0 license. 

Additional Editor Comments:

Line 84: Change "outdoor-raised domestic pig premises" to "OPO" unless there its a difference between OPOs and a "premise" which does not seem the case.

Line 125: Change "Publicly available predictor spatial layers online" to "Predictor spatial layers were publicly available online." The authors often and excessively use adjective modifiers throughout their writing and should avoid if possible. Using 3-5 adjectives to describe your subject gets confusing to interpret exact meaning of your topic.

Line 149: Change "1" to "one" or delete altogether? Not clear if you are stating that the default was determined to be the optimal setting so is 1 even needed?

Line 262: Change "outdoor-raised pig operations" to "OPOs."

Line 264-265: While I believe this is appropriate language for a Cover Letter, I suggest the authors delete this here because you say nearly the same thing in the subsequent sentence. Also, remove "to our knowledge" in the Lines 265-267.

Lines 268-269: Change "species distribution predictive methods" to "predictive methods for species distribution."

Line 270: remove "MaxEnt" because it is not necessary.

Line 275: Again remove "MaxEnt" throughout Discussion. You state the models run in your Methods so no reason to refer to MaxEnt each time you reference your models or results in the Discussion.

Line 278: Change "the final MaxEnt model was" to "our results were"

Line 284: NAHMS? Spell out.

Reviewers' comments:

Reviewer's Responses to Questions

**Comments to the Author**

1. Is the manuscript technically sound, and do the data support the conclusions?

Reviewer #1: Yes

Reviewer #2: Yes

2. Has the statistical analysis been performed appropriately and rigorously? 

Reviewer #1: Yes

Reviewer #2: I Don't Know

3. Have the authors made all data underlying the findings in their manuscript fully available?

Reviewer #1: Yes

Reviewer #2: Yes

4. Is the manuscript presented in an intelligible fashion and written in standard English?

Reviewer #1: Yes

Reviewer #2: Yes

5. Review Comments to the Author

Reviewer #1: Comments

--------

In this study, Patterson et al. use hunter-harvested pigs and landscape

variables to develop an occupancy map of feral swine in California. They then

use farmer surveys to identify outdoor pig operations and overlay their

occupancy and OPO map to predict areas of high contact risk between feral swine

and domestic swine. The study provides one of the first high resolution maps

of potential contact zones between domestic and feral pigs in California.

Overall, I found the results and conclusions sound and consistent with previous

analyses on feral swine resource selection and occupancy.

My main comment is that I think more discussion should be provided on

the "implications for disease transmission" portion of this manuscript. The

bulk of the discussion (6 paragraphs) focuses on the MaxEnt results while the

discussion of transmission implications are given in 2 paragraphs. The MaxEnt

results are largely consistent with previous analyses on feral swine and, given

the paper's title, I think there should be more balance between predictors of

pig relative occupancy and transmission implications. For example, beyond

actual disease data, what are some additional components of transmission that

the proposed measure of contact risk is lacking? At least three come to mind:

i) this measure is not accounting for the size of OPOs (though it looks like

this was measured in the surveys) ii) the measure is not accounting for the

density of feral pigs in an area and iii) the measure is not accounting for the

connectivity of the OPOs (mentioned in my minor comments below). All of these

factors could significantly alter how contact risk translates to transmission

risk and spread among farms. Balancing the discussion with a more in-depth

look at the transmission implications of these results would strengthen the

paper.

Minor Comments

--------------

Line 77: What about Lewis et al. "Historical, current, and potential population

size estimates of invasive wild pigs (Sus scrofa) in the United States" who map

feral swine density at the 1 km scale?

Line 130: Change "NDIV" to "NDVI"

Line 138: This is a bit vague. What does "maintaining adequate detail for

suitable habitat modeling" mean? For example, could you clarify why a 1 km x 1

km scale would have been inadequate for the goals of this study?

Table 1: Please provide the spatial resolution of the layers (where applicable)

Line 159: Change "was" to "were"

Figure 1: It would be useful if the author's could overlay a map of the observed

hunter harvest points.

Line 228: Change "OPO" to "OPOs"

Results: I could not figure out how to access the Supplementary Material so I

was unable to review the response curves and S1 Table, S1 Fig, and S2 Fig.

The authors may have been limited by space, but if possible it would be

useful to see the response curves in the main text.

Figure 2: Colors on this figure are hard to see. Also, I don't understand the

color bar label. Why is Lowest Risk greater than (>) Moderate Risk greater

than (>) Highest Risk?

Line 265: Pepin et al. 2021 (Prev. Vet. Medicine) also does this, but at the

county scale and crossing the wildlife-livestock-human interface.

Line 276-277: But don't you have the observed point so you know whether or not

feral swine were observed in these counties? The "may indicate" is confusing.

Perhaps "reflects"?

Line 335-336: Could you remind the reader what criteria you are using to make

this assessment. Also, if I am interpreting this sentence correctly, it would

be helpful to rephrase similar to the following: "Additionally, using a model

fitted with 2017 hunting tags (n=1,745) vs. all 5,148 points for 2012-19

provided the best out-of-sample predictions".

Line 361-362, 415-417: It would be pertinent to mention that farms with the

highest spillover risk might not be the most connected to other farms and might

be less important for among-farm spread than suggested by the maps given here.

Mitigation might be most effective if highly connected farms are targeted even

if they have lower feral swine suitability.

Line 373-374: I don't understand how the "therefore" clause follows in this

sentence. I would recommend re-wording this sentence to make it more clear.

Reviewer #2: Overall this manuscript looks great! I think it is a nice body of work that is translational and has true One Health implications. I have provided additional comments in the manuscript but I do think this manuscript is worthy of acceptance.

6. PLOS authors have the option to publish the peer review history of their article (what does this mean?). If published, this will include your full peer review and any attached files.

Reviewer #1: No

Reviewer #2: No

---

## [Author Response · Author response to Decision Letter 0]

23 May 2022

PONE-D-21-36324

Identification of high-risk contact areas between feral pigs and outdoor-raised pig operations in California: implications for disease transmission in the wildlife-livestock interface

PLOS ONE

Editor Comments: 

I agree with the comments by Reviewer 1 but I believe that Major Revision is more appropriate. I don't believe the level of detail of the dataset and spatial layers is adequate. In their revision I would appreciate the authors including details for their methods outlined below:

AR1: The authors really appreciate the comments and suggestions of the reviewers. The authors have done their best to address all of the points addressed by the editor and reviewers. We updated the tables and figures and added two more figures, including the feral pig hunting tags. The Journal requirements were followed and updated. Additionally, text regarding disease transmission and anthropogenic movement was added to the discussion section.

1. As mentioned by Reviewer 1, the level of detail of how each spatial layer was presented into MaxEnt is not adequate. I needed to see Figure 1 before realizing that this was throughout the state of California and not a portion of it. Table 1 could include resolution of each layer and how each layer was presented. The level of detail in Description column is not nearly enough. I would suggest moving this to the Methods text and add more details. I don't believe anyone could replicate your models with this level of detail for your spatial layers.

AR2: Thank you for raising this concern. All layers' initial resolutions and projections are available at each website source presented in Table 1. For the purpose of this study, we reprojected all layers to the Albers Equal-Area coordinate reference system for California (“California Albers” (meters)) using QGIS 3.6. If a layer is not specific to California, we cropped it to fit specifically to the extent of California and added a line explaining that layers were masked for entire state of California: “and masked for the entire state of California”. Lines 142-143.

Maxent requires that all layers have the same size (pixel size) which we fixed to 270 x 270 m. This has been clarified in the text and Table 1 to facilitate replication. Additionally, we have included the standardized five layers that were significant in the final model in the Supplementary section. More details about this modeling approach can be found in Hijmans RJ, Phillips S, And JL, Elith A. dismo: Species Distribution Modeling. R package version 1.3-3. 2019. Additionally, we added more text to the Materials and Methods section to clarify descriptions of the main layers used in the final MaxEnt model. See Table 1 and Lines 130-136; 142-147.

2. Was AVGMODIS an average of 12 years of NDVI.

AR3: Yes, AVGMODIS is an average of 12 years of NDVI as is written here: “For example, AVGMODIS was the annual maximum green vegetation fraction (MGVF) combined with 12 years of normalized difference vegetation index data (NDVI) and relates to food and shrub cover for feral pigs.” We added this text to clarify about this layer: NDVI measures vegetation gathered by the Moderate Resolution Imaging Spectroradiometer (MODIS) as part of NASA’s satellite systems. Lines 134-136.

3. If AVGMODIS is NDVI is it correlated to NDVI? Were any correlations included in your data sources prior to using them in MaxEnt?

AR4: Thank you for the suggestion. We did run correlations and I added in this text to clarify: “Predictors were assessed for correlation using Spearman’s rank and a cut-off threshold of 0.80, a threshold used in previous studies (LaHue et al, 2016). Two correlated variables were not included at the same time, during variable selection steps.” Lines 145-147.

4. What year was NLCD used as a spatial layer? NLCD is released every 5 years so which version should be placed in the methods.

AR5: NCLD layer used was from 2016. We added a ‘Years’ column to Table 1.

5. As requested by Reviewer 1, feral pig tag locations across the state needs to be included if the authors are presenting a risk map statewide.

AR6: We have included a map containing feral pig hunting tags from 2017. MaxEnt uses a variety of layers to predict suitable habit, along with presence points, so overlaying hunting tag points is contrary to the purpose of using MaxEnt. Hunting tags represent a subset of feral pig locations, because legal hunting is only conducted on public lands, which encompasses approximately 50% of California. 

6. Was the OPO survey done as part of this study? The response rate of these surveys is important to report. Do the authors believe their survey adequately reflects OPOs (13 percent response rate) in California based on their results? It seems the survey questions should be a supplement or a separate manuscript? If the authors have coordinates for OPOs but only a 13% response rate, is the survey of any value in this manuscript?

AR7: Yes, the survey was conducted to gather the locations of OPO and investigate feral pig presence nearby these type of operations. We added this text to clarify: ‘Additionally as part of our objective to gather locations of OPOs in California…’ Line 186.

The survey only had 44 respondents, because of the recruitment strategy (snow-ball approach) and there is not a census of those operations, therefore, we are not able to estimate a response rate. See this text within the manuscript: “This online survey was announced electronically (e.g., media, e-newsletters) to swine related groups and organizations or conducted in-person at events, such as agricultural fairs.” Lines 191-193.

Regarding whether we “believe their survey adequately reflects OPOs (13 percent response rate) in California, farms are not required to register with any government agency, therefore no one really knows the number of OPOs nationwide, although the USDA has been editing their national surveys over time to include operations that raise swine outdoors. Please see this text that was in the limitations section “For instance, farms and ranches in California, including backyard and commercial operations, are not required to register with state agricultural agencies, therefore, the total number, distribution, and size of OPO remains unknown and are underrepresented in this study. A majority of the identified OPO in this study were commercial pork producers with an online presence or ones that attend conferences, farmers markets and fairs.” Lines 460-465.

The authors believe it is important to include this survey because 1) it demonstrates how challenging it is to identify OPOs and 2) although a small number of famers participated, it provides demographic details regarding this population that is not reflected in mapping locations only.

Journal Requirements:

1. Manuscript Meets PLOS One’s style requirements. 

2. Financial Disclosure

We note that the grant information you provided in the ‘Funding Information’ and ‘Financial Disclosure’ sections do not match

3. Data Availability: 

In your Data Availability statement, you have not specified where the minimal data set underlying the results described in your manuscript can be found.

4. We note that Figures 1 and 2 in your submission contain map images which may be copyrighted. 

AR8: The manuscript and files were revised and modified to meet the style requirements. Financial disclosure and funding information were corrected. 

Funding: This study was supported by the Agriculture and Food Research Initiative grant no. 2019-67011-29609 from the USDA National Institute of Food and Agriculture (LP) and a University of California, Davis Academic Federation grant (LP and AFAP). The funders had no role in study design, data collection and analysis, decision to publish, or preparation of the manuscript.

Data statement was modified to: OPO data cannot be shared publicly, as they contain GPS coordinates and participants provided locations based on confidentiality. Feral pig hunting tags are provided in the text as Fig1. Predictor layers used to build a MaxEnt model are publicly available online, but rasters included in the final MaxEnt model are included in KML format as supplementary files. 

Figures: Figure 1 (Hunting tags), Figure 2 (Maximum Entropy model that predicts suitable feral pig habitat) and Figure 4 (Risk map) were all created by the first author (i.e., originals) and are not copied nor subjected to copyrights.

Additional Editor Comments:

Line 84: Change "outdoor-raised domestic pig premises" to "OPO" unless there its a difference between OPOs and a "premise" which does not seem the case. 

AR9: Suggestion accepted. Line 84.

Line 125: Change "Publicly available predictor spatial layers online" to "Predictor spatial layers were publicly available online." The authors often and excessively use adjective modifiers throughout their writing and should avoid if possible. Using 3-5 adjectives to describe your subject gets confusing to interpret exact meaning of your topic. 

AR10: Suggestion accepted. Line 128.

Line 149: Change "1" to "one" or delete altogether? Not clear if you are stating that the default was determined to be the optimal setting so is 1 even needed? 

AR11: Correction was made. Line 157-168.

Line 262: Change "outdoor-raised pig operations" to "OPOs." 

AR12: Suggestion accepted. Line 263 and throughout the manuscript (see yellow highlights).

Line 264-265: While I believe this is appropriate language for a Cover Letter, I suggest the authors delete this here because you say nearly the same thing in the subsequent sentence. Also, remove "to our knowledge" in the Lines 265-267. 

AR13: Thank you for the suggestion. The first paragraph of discussion was modified. Lines 281-289

Lines 268-269: Change "species distribution predictive methods" to "predictive methods for species distribution." 

AR14: Suggestion accepted. Lines 290-291.

Line 270: remove "MaxEnt" because it is not necessary. 

AR15: Suggestion accepted. Lines 292-293.

Line 275: Again remove "MaxEnt" throughout Discussion. You state the models run in your Methods so no reason to refer to MaxEnt each time you reference your models or results in the Discussion. 

AR16: Thank you for the suggestion, we removed “MaxEnt” throughout the discussion.

Line 278: Change "the final MaxEnt model was" to "our results were". 

AR17: Suggestion accepted. Lines 299

Line 284: NAHMS? Spell out. 

AR18: Suggestion accepted. Lines 305-36.

Comments to the Author

Reviewer #1: Comments

In this study, Patterson et al. use hunter-harvested pigs and landscape variables to develop an occupancy map of feral swine in California. They then use farmer surveys to identify outdoor pig operations and overlay their occupancy and OPO map to predict areas of high contact risk between feral swine and domestic swine. The study provides one of the first high resolution maps of potential contact zones between domestic and feral pigs in California. Overall, I found the results and conclusions sound and consistent with previous

analyses on feral swine resource selection and occupancy.

My main comment is that I think more discussion should be provided on the "implications for disease transmission" portion of this manuscript. The bulk of the discussion (6 paragraphs) focuses on the MaxEnt results while the discussion of transmission implications are given in 2 paragraphs. The MaxEnt results are largely consistent with previous analyses on feral swine and, given the paper's title, I think there should be more balance between predictors of pig relative occupancy and transmission implications. For example, beyond actual disease data, what are some additional components of transmission that the proposed measure of contact risk is lacking? At least three come to mind: i) this measure is not accounting for the size of OPOs (though it looks like this was measured in the surveys) ii) the measure is not accounting for the density of feral pigs in an area and iii) the measure is not accounting for the connectivity of the OPOs (mentioned in my minor comments below). All of these factors could significantly alter how contact risk translates to transmission risk and spread among farms. Balancing the discussion with a more in-depth look at the transmission implications of these results would strengthen the paper.

AR18: The authors appreciated the suggestion of the reviewer. We added a section in the discussion section regarding implications for disease transmission. Lines 398-428 

Minor Comments

Line 77: What about Lewis et al. "Historical, current, and potential population size estimates of invasive wild pigs (Sus scrofa) in the United States" who map feral swine density at the 1 km scale?

AR19: The reviewer raised a very good point. Lewis et al, measured at a 1km scale, but they were trying to estimate the abundance of feral pigs per 1km x 1km cell size for the entire US, whereas we were identifying possible areas of contact between a feral pig and OPOs, which is a smaller scale level. (See lines 102-104 “To the best of our knowledge, there are no maps characterizing where suitable feral pig habitat overlaps with domestic pigs raised outdoors at the farm-level in California.”) Additionally, because they were running models for the entire nation, using a smaller pixel size (as we did in the current study) would have taken a lot of computer processing time. Because we were working only with one state, California, we could run models with smaller cell sizes, which provided a finer scale prediction per 270m x 270m area. We modified the introduction to: ‘Feral pig population distribution and abundance is dynamic yet has not been documented at fine spatial units less than 1km. Additionally, previous presence maps reported feral pigs for an entire county, even if there had only been a single occurrence recorded countywide’ Lines 76-79.

Line 130: Change "NDIV" to "NDVI"

AR20: Correction made. Line 134.

Line 138: This is a bit vague. What does "maintaining adequate detail for suitable habitat modeling" mean? For example, could you clarify why a 1 km x 1 km scale would have been inadequate for the goals of this study?

AR21: The OPO are generally small-scaled and since we were modeling the interface at the farm-level, we wanted to use a resolution that matched closer to a small farm size (270 meter vs 1 km). We changed the text to “Rasters were all converted to the same resolution of 270m x 270m, which used a reasonable amount of computer computation time, while maintaining fine-scale for suitable habitat modeling at the farm-level”. Lines 143-145.

Table 1: Please provide the spatial resolution of the layers (where applicable)

AR22: Suggestion accepted. We adjusted Table 1 to include years and original layer resolution.

Line 159: Change "was" to "were".

AR23: Correction made. Line 153

Figure 1: It would be useful if the author's could overlay a map of the observed hunter harvest points. 

AR24: Thank you for the suggestion. We included a map of 2017 hunting tags. Figure 1. 

Line 228: Change "OPO" to "OPOs" . 

AR25: Corrected. Line 248.

Results: I could not figure out how to access the Supplementary Material so I was unable to review the response curves and S1 Table, S1 Fig, and S2 Fig. The authors may have been limited by space, but if possible it would be useful to see the response curves in the main text.

AR26: Thank you for the suggestion. We added the response curves to the main document. Lines 240-245. Figure 3.

Figure 2: Colors on this figure are hard to see. Also, I don't understand the color bar label. Why is Lowest Risk greater than (>) Moderate Risk greater than (>) Highest Risk?

AR26: Thank you for the suggestion. We re-formatted the figure and legend of the Risk map. The background of the risk map now matches the legend better and the legend was fixed to remove the “>”. Figure 4

Line 265: Pepin et al. 2021 (Prev. Vet. Medicine) also does this, but at the county scale and crossing the wildlife-livestock-human interface.

AR27: This is a good point raised by the reviewer. We updated the first paragraph of the discussion section. See Lines 281-289. 

Line 276-277: But don't you have the observed point so you know whether or not feral swine were observed in these counties? The "may indicate" is confusing. Perhaps "reflects"?

AR28: This is a good point. We changed to “reflects” Line 258. Hunting tags contain bias, because hunting is only conducted on public lands and not in all counties. Additionally, not all hunters report GPS locations accurately. This limitation is described in Lines 440-444.

Line 335-336: Could you remind the reader what criteria you are using to make this assessment. Also, if I am interpreting this sentence correctly, it would be helpful to rephrase similar to the following: "Additionally, using a model fitted with 2017 hunting tags (n=1,745) vs. all 5,148 points for 2012-19 provided the best out-of-sample predictions".

AR29: Thank you for the suggestion. We rephrased this sentence and added a new sentence explaining the AUC assessment, in Materials Methods section. “In general, the following guidelines are used to assess models with AUC: 0.90 and above indicates an excellent model, a good model ranges from 0.80-0.90, a fair model runs between 0.70-0.80 and poor or failed model is any value under 0.70.” Lines 171-173 and 362-363.

Line 361-362, 415-417: It would be pertinent to mention that farms with the highest spillover risk might not be the most connected to other farms and might be less important for among-farm spread than suggested by the maps given here. Mitigation might be most effective if highly connected farms are targeted even if they have lower feral swine suitability.

AR30: This is a good point. We added a new sentence to the discussion: “Targeting outreach and surveillance to highly connected farms may be warranted, even if that farm is not near highly suitable feral pig habitat, given their effect on other farms. However, OPO with the highest risk of disease spillover may not be connected to other OPOs.” Lines 391-394.

Line 373-374: I don't understand how the "therefore" clause follows in this

sentence. I would recommend re-wording this sentence to make it more clear. 

AR31: This sentence was removed.

Reviewer #2: Comments

Overall I think this manuscript is a really nice body of work. There is a lot of work done on the feral swine side, a lot of work done on the domestic swine side, and a lot of work done on the human health side but you don't often see comprehensive studies so I really appreciated that. One important consideration that I think needs to be included in the discussion is anthropogenic movement. Feral swine are routinely translocated via human-mediated movement which is very different than most other wildlife species. I think that is an important consideration when thinking about environmental conditions and warrants discussion. Great job!

AR32: Thank you for the positive feedback. We added a paragraph about anthropogenic movement in the discussion section. Lines 359-367.

Other minor changes: 

Line 71: ‘as’ =‘has’. AR33: Changed. Line 72

Line 122: Capitalized ‘methods’ to Methods. AR34: Changed Line 112

Line 155: Add ‘were’. AR35: Changed. Line 167

Line 195: Change ‘sex’ for ‘gender’. AR35: Changed. Line 210

---

## [Editor Report · Decision Letter 1]

13 Jun 2022

Identification of high-risk contact areas between feral pigs and outdoor-raised pig operations in California: implications for disease transmission in the wildlife-livestock interface

PONE-D-21-36324R1

Dear Dr. Pires,

We’re pleased to inform you that your manuscript has been judged scientifically suitable for publication and will be formally accepted for publication once it meets all outstanding technical requirements.

Kind regards,

W. David Walter, Ph.D.

Academic Editor

PLOS ONE

Additional Editor Comments (optional):

I appreciate the authors addressing my comments as well as those of the reviewers. I believe the level of detail added to this draft by the authors provided sufficient information to your Methods, spatial layers, and MaxEnt models that was requested by reviewers.
---

## [Editor Report · Acceptance letter]

20 Jun 2022

PONE-D-21-36324R1 

Identification of high-risk contact areas between feral pigs and outdoor-raised pig operations in California: implications for disease transmission in the wildlife-livestock interface 

Dear Dr. Pires:

I'm pleased to inform you that your manuscript has been deemed suitable for publication in PLOS ONE. Congratulations! Your manuscript is now with our production department. 

Kind regards, 

on behalf of

Dr. W. David Walter 

Academic Editor

PLOS ONE